# Salivary molecular spectroscopy: A sustainable, rapid and non-invasive monitoring tool for diabetes mellitus during insulin treatment

Douglas C. Caixeta[1,2], Emília M. G. Aguiar[1], Léia Cardoso-Sousa[1], Líris M. D. Coelho[1], Stephanie W. Oliveira[1], Foued S. Espindola[2], Leandro Raniero[3], Karla T. B. Crosara[4], Matthew J. Baker[5], Walter L. Siqueira[4], Robinson Sabino-Silva[1,4]*

1 Department of Physiology, Institute of Biomedical Sciences, Federal University of Uberlandia, Uberlandia, Minas Gerais, Brazil, 2 Institute of Genetics and Biochemistry, Federal University of Uberlandia, Uberlandia, Minas Gerais, Brazil, 3 Nanosensor Laboratory, IP&D, University of Vale do Paraíba, São José Dos Campos, SP, Brazil, 4 College of Dentistry, University of Saskatchewan, Saskatoon, Saskatchewan, Canada, 5 WestCHEM, Department of Pure and Applied Chemistry, Technology & Innovation Centre, University of Strathclyde, Glasgow, United Kingdom

* robinsonsabino@gmail.com

**Data Availability Statement:** All relevant data are within the manuscript and its Supporting Information files.

## Abstract

Monitoring of blood glucose is an invasive, painful and costly practice in diabetes. Consequently, the search for a more cost-effective (reagent-free), non-invasive and specific diabetes monitoring method is of great interest. Attenuated total reflectance Fourier transform infrared (ATR-FTIR) spectroscopy has been used in diagnosis of several diseases, however, applications in the monitoring of diabetic treatment are just beginning to emerge. Here, we used ATR-FTIR spectroscopy to evaluate saliva of non-diabetic (ND), diabetic (D) and insulin-treated diabetic (D+I) rats to identify potential salivary biomarkers related to glucose monitoring. The spectrum of saliva of ND, D and D+I rats displayed several unique vibrational modes and from these, two vibrational modes were pre-validated as potential diagnostic biomarkers by ROC curve analysis with significant correlation with glycemia. Compared to the ND and D+I rats, classification of D rats was achieved with a sensitivity of 100%, and an average specificity of 93.33% and 100% using bands 1452 cm⁻¹ and 836 cm⁻¹, respectively. Moreover, 1452 cm⁻¹ and 836 cm⁻¹ spectral bands proved to be robust spectral biomarkers and highly correlated with glycemia ($R^2$ of 0.801 and 0.788, P < 0.01, respectively). Both PCA-LDA and HCA classifications achieved an accuracy of 95.2%. Spectral salivary biomarkers discovered using univariate and multivariate analysis may provide a novel robust alternative for diabetes monitoring using a non-invasive and green technology.

## Introduction

Diabetes mellitus (DM) is a metabolic disorder characterized by hyperglycemia which results from insufficient secretion and/or reduced insulin action in peripheral tissues [1, 2]. According to the International Diabetes Federation (IDF), there are an estimated 425 million adults

**Funding:** This research was supported by a grant from CAPES/CNPq (#458143/2014), FAPEMIG (#APQ-02872-16), Federal University of Uberlandia and National Institute of Science and Technology in Theranostics and Nanobiotechnology (CNPq Process N.: 465669/2014-0). Canadian Institutes of Health Research (CIHR grants #106657 and # 400347). CAIXETA, D.C.; AGUIAR, E. M. G.; and CARDOSO-SOUSA, L. received a fellowship from FAPEMIG, CNPq e CAPES, respectively. Sabino-Silva, R received a fellowship from PrInt CAPES/UFU. We would like to thank our collaborators at the Rodent Vivarium Network (REBIR-UFU) and Dental Research Center in Biomechanics, Biomaterials and Cell Biology (CPbio-UFU). The funders had no role in study design, data collection and analysis, decision to publish, or preparation of the manuscript.

**Competing interests:** No competing interests.

with diabetes worldwide, these include 212 million who are estimated undiagnosed [3]. Frequent monitoring of diabetes is essential for improved glucose control and to delay clinical complications related with diabetes. Besides, the early screening of DM is paramount to reduce the complications of this metabolic disorder worldwide [4]. Despite being relatively invasive and painful, blood analysis per glucometer is currently feasible for screening, monitoring and diagnosing diabetes by needle finger punctures [5, 6]. The constant need of piercing the fingers several times daily by most patients is inconvenient and may lead to the development of finger calluses and difficulty in obtaining blood samples [5].

Saliva reflects several physiological functions of the body [7, 8]. In this way, salivary biomarkers might be an attractive alternative to blood for early detection, and for monitoring systemic diseases [9]. Among the advantages, saliva is simple to collect, non-invasive, convenient to store and, compared to blood, requires less handling during clinical procedures. Besides, saliva also contains analytes with real-time monitoring value which can be used to check the individuals condition [8, 10]. Currently, a broad set of methods are used to analyze saliva including immunoassays, colorimetric, enzymatic, kinetic, chromatographic and mass spectrometric analysis [11]. Several studies showed higher salivary glucose levels in DM patients than non-hyperglycemic controls, however, the studies reject the idea of a direct relationship between salivary glucose and glycemia in diabetic patients [6, 12–17]. Another limitation of salivary-based measurement of glucose for diabetes monitoring is the presence of glucose in foods, which can disturb the monitoring process as it induces changes in salivary glucose concentration. Therefore, other alternatives of salivary monitoring should be studied.

Infrared (IR) spectroscopy is emerging as a powerful quantitative and qualitative technique for monitoring characterization of biological molecules in fluids [18]. Attenuated total reflection Fourier-transform infrared (ATR-FTIR) spectroscopy is a global, sensitive and highly reproducible physicochemical analytical technique that identifies structural molecules on the basis of their IR absorption [19]. Considering that a biomolecule is determined by its unique structure, each one will exhibit a unique ATR-FTIR spectrum, representing the vibrational modes of the constituent structural bonds [19, 20]. ATR-FTIR is a green technology due to processes that eliminate the use of hazardous elements an overarching approach that is applicable to monitoring diseases. The IR spectral modes of biological samples, such as saliva, may be considered as biochemical fingerprints that correlate directly with the presence or absence of diseases, and, furthermore, provide the basis for the quantitative determination of several analytes for monitoring several diseases and to diagnostic interest [21, 22]. The potential of salivary diagnostic for diabetes by IR spectroscopy using barium fluoride ($BaF_2$) slides was suggested previously [23], however, the efficacy of DM monitoring in insulin-treated conditions using ultra-low volumes of saliva remains unknown.

In the present study, we tested the hypothesis that non-invasive spectral biomarkers can be identified in saliva of hyperglycemic diabetic and in insulin-treated diabetic rats, and the differentially expressed vibrational modes can be employed as salivary biomarkers for diabetes monitoring. Thus, the aim of our study was to identify infrared spectral signatures of saliva that are suitable to monitoring this metabolic disease in untreated and insulin-treated conditions. For this, the salivary vibrational modes profile of non-diabetic, diabetic and insulin-treated diabetic rats was quantitatively and qualitatively evaluated using univariate and multivariate analysis.

## Results

### Characterization of diabetes mellitus

To confirm the effectiveness of diabetes induction and insulin treatment, several parameters were assessed in anesthetized animals. As expected, to confirm the diabetic state, Table 1

**Table 1. Effect of diabetes and insulin on body weight, water intake, food intake, glycemia, urine volume and urine glucose concentration.**

| Parameters | ND | D | D+I |
|---|---|---|---|
| Δ Body weight (g) | 48.4±8.3 | -2.7±11.3* | 39.5±12.8# |
| Water intake (mL) | 39.1±3.1 | 150.6±17.9* | 60.0±6.8# |
| Food intake (g) | 18.3±1.3 | 35.0±4.1* | 29.7±2.6* |
| Glycemia (mg/dL) | 83.2±4.2 | 497.6±19.6* | 81.0±19.2# |
| Urine volume (mL) | 22.1±3.4 | 128.9±8.6* | 40.7±7.1# |
| Urine glucose (mg/dL) | 24.7±7.2 | 337.2±15.8* | 148.0±34.6*# |

*$p < 0.05$ vs ND

#$P < 0.05$ vs D; one-way ANOVA followed by Student Newman Keuls post-test.

shows that diabetes reduced weight gain ($p < 0.05$), increased water intake ($p < 0.05$) and food ingestion ($p < 0.05$) compared with ND rats. Besides, in diabetic condition, higher plasma glucose ($p < 0.05$), as well as most pronounced urine volume ($p < 0.05$), associated with higher urine glucose concentration ($p < 0.05$), were observed in D rats compared with ND rats. Insulin treatment contributed to increased ($p < 0.05$) weight gain and decreased water intake ($p < 0.05$) compared with placebo-treated D rats. As expected, insulin treatment decreased plasma glucose ($p < 0.05$), urine volume ($p < 0.05$) and urine glucose concentration compared with D rats. Glycemia and urine volume were similar ($p > 0.05$) in ND and D+I animals, indicating that insulin treatment completely reverted hyperglycemia and higher urine volume described in D rats. The insulin treatment promoted a strong reduction in the urinary glucose concentration; however, the urinary glucose concentration was increased ($p < 0.05$) in D+I compared to ND animals.

## Average spectra of saliva

A representative infrared average spectrum of saliva from normoglycemic, hyperglycemic and insulin-treated conditions, which contains different molecules such as lipids, proteins, glycoproteins and nucleic acid, are represented in Fig 1A. These salivary spectra indicated several differences among non-diabetic, diabetic and insulin-treated diabetic rats. Some bands of interest are shown in Fig 1, which contains: asymmetric stretching vibration of $CH_2$ of acyl chains of lipids (2924 cm$^{-1}$); amide II (1549 cm$^{-1}$); asymmetric $CH_3$ bending modes of the methyl groups of proteins (1452 cm$^{-1}$); amide III band components of proteins (1313 cm$^{-1}$); mannose-6-phosphate and phosphorylated saccharide residue (1120 cm$^{-1}$) and $C_2$ conformation of sugar (836 cm$^{-1}$). The representative spectral changes compared to ND rats was represented in Fig 1B.

## Spectral bands analyzed by IR spectroscopy

Spectral band areas that indicate the expression of specific molecules were analyzed in saliva. The band area values of 2924 cm$^{-1}$, 1549 cm$^{-1}$, 1313 cm$^{-1}$, 1120 cm$^{-1}$ are presented in supplementary files. Herein, we showed two bands (1452 cm$^{-1}$ and 836 cm$^{-}$1) with a higher potential for diabetes monitoring (Fig 2 and Fig 3, respectively). Representative spectra of 1452 cm$^{-1}$ and 836 cm$^{-}$1 bands are depicted in Figs 2A and 3A. Diabetes induced a decrease ($p < 0.05$) at 1452 cm$^{-1}$ and 836 cm$^{-1}$ bands compared with non-diabetic rats, however, insulin-treated diabetic reverted this alteration in both bands (Figs 2A and 3A, respectively).

To investigate whether these salivary vibrational modes would be reflective of glycemia regulation, these two salivary band areas were discovered to be, via univariate analysis, the best

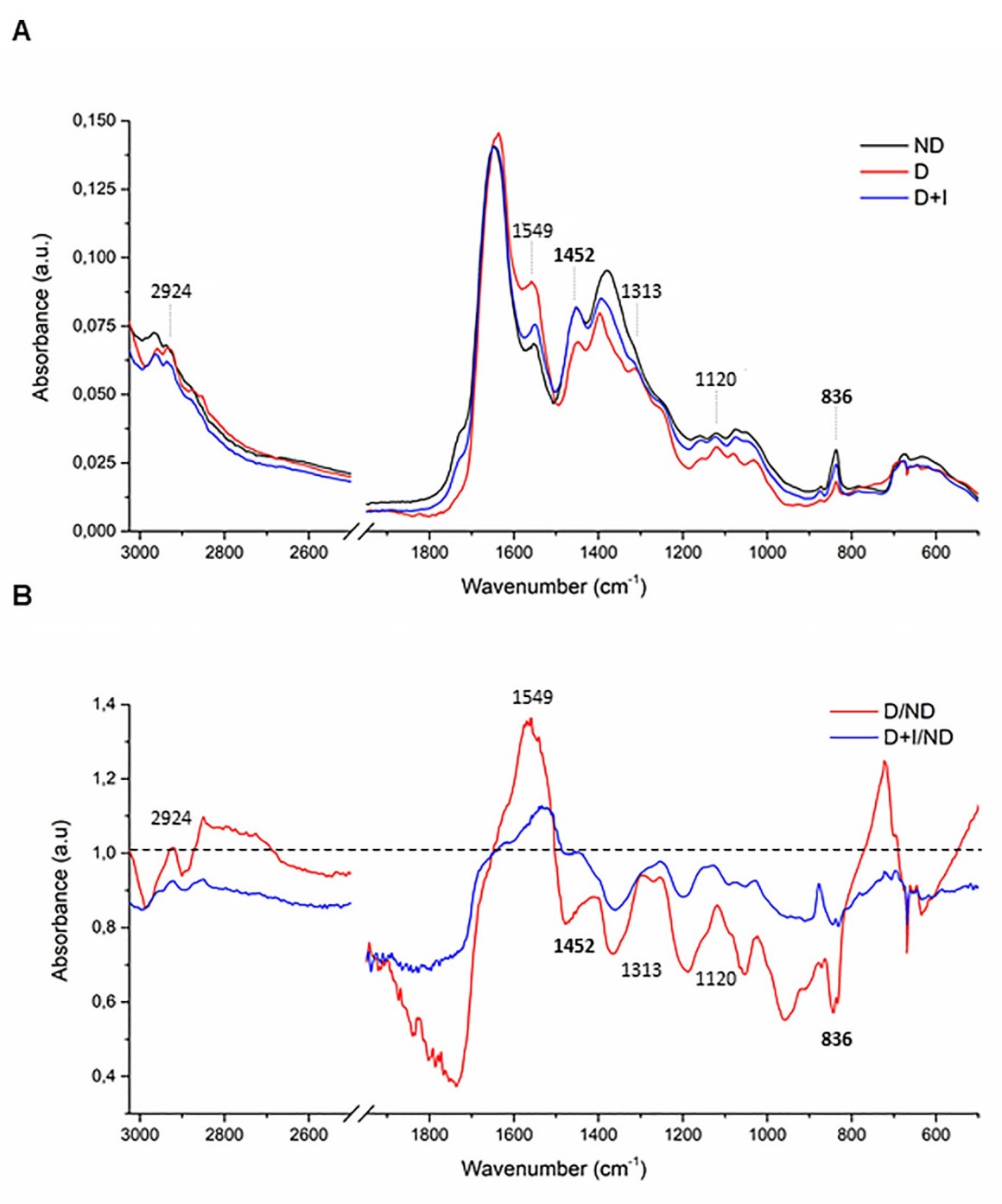

**Fig 1.** (A) Representative average ATR-FTIR spectra (3000–400 cm-1) in saliva of Non-Diabetic rats (ND), diabetic rats (D) and diabetic treated with insulin (D+I). (B) Representative spectral changes compared to ND rats.

spectral candidates values to indicate the diabetes monitoring in samples with hyperglycemia, normoglycemia and under insulin treatment. Pearson's correlation between these spectral modes (1452 cm$^{-1}$ and 836 cm$^{-1}$) with glycemia showed high correlation. The both salivary spectral bands presented strong negative correlation with r = -0.801; p < 0.0001 for 1452 cm$^{-1}$ (Fig 2C) and r = -0.788; p < 0.0001 for 836 cm$^{-1}$ (Fig 3C).

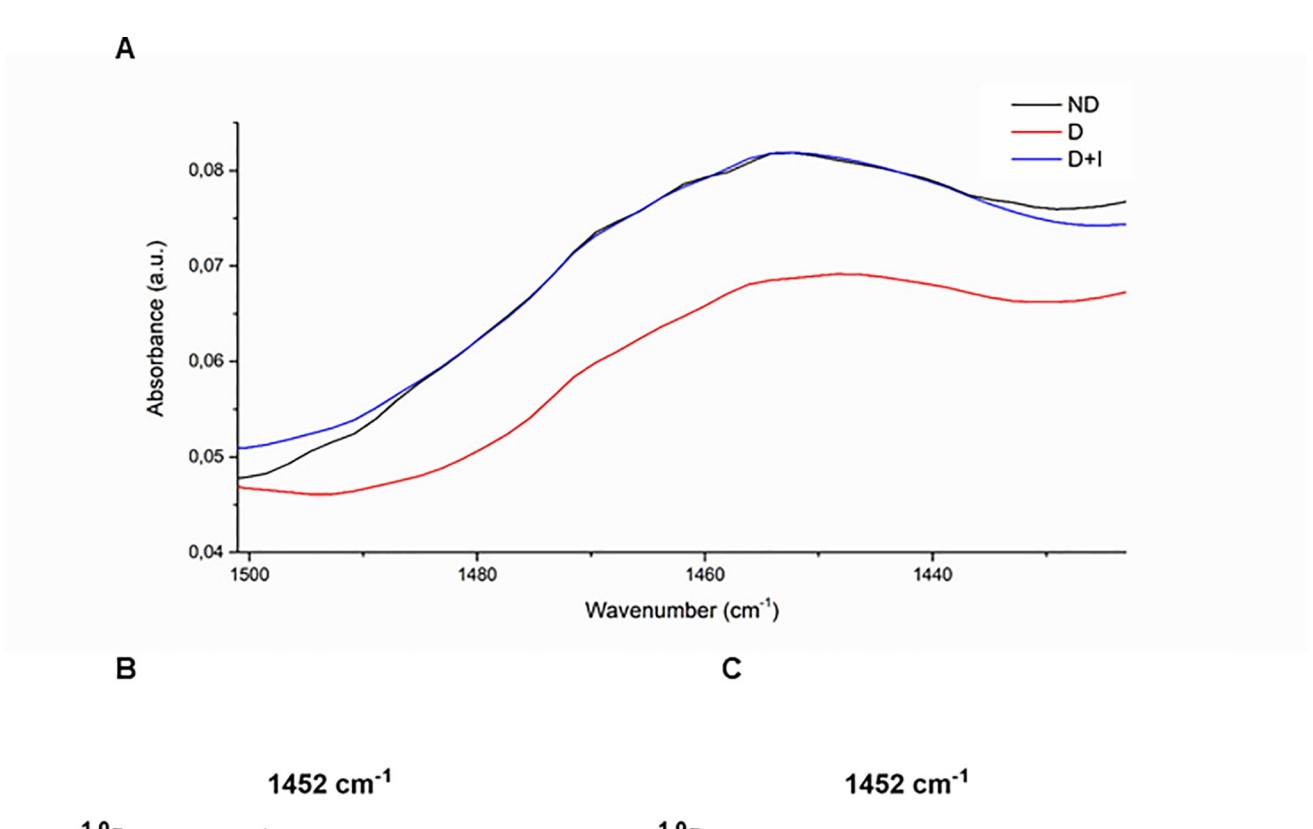

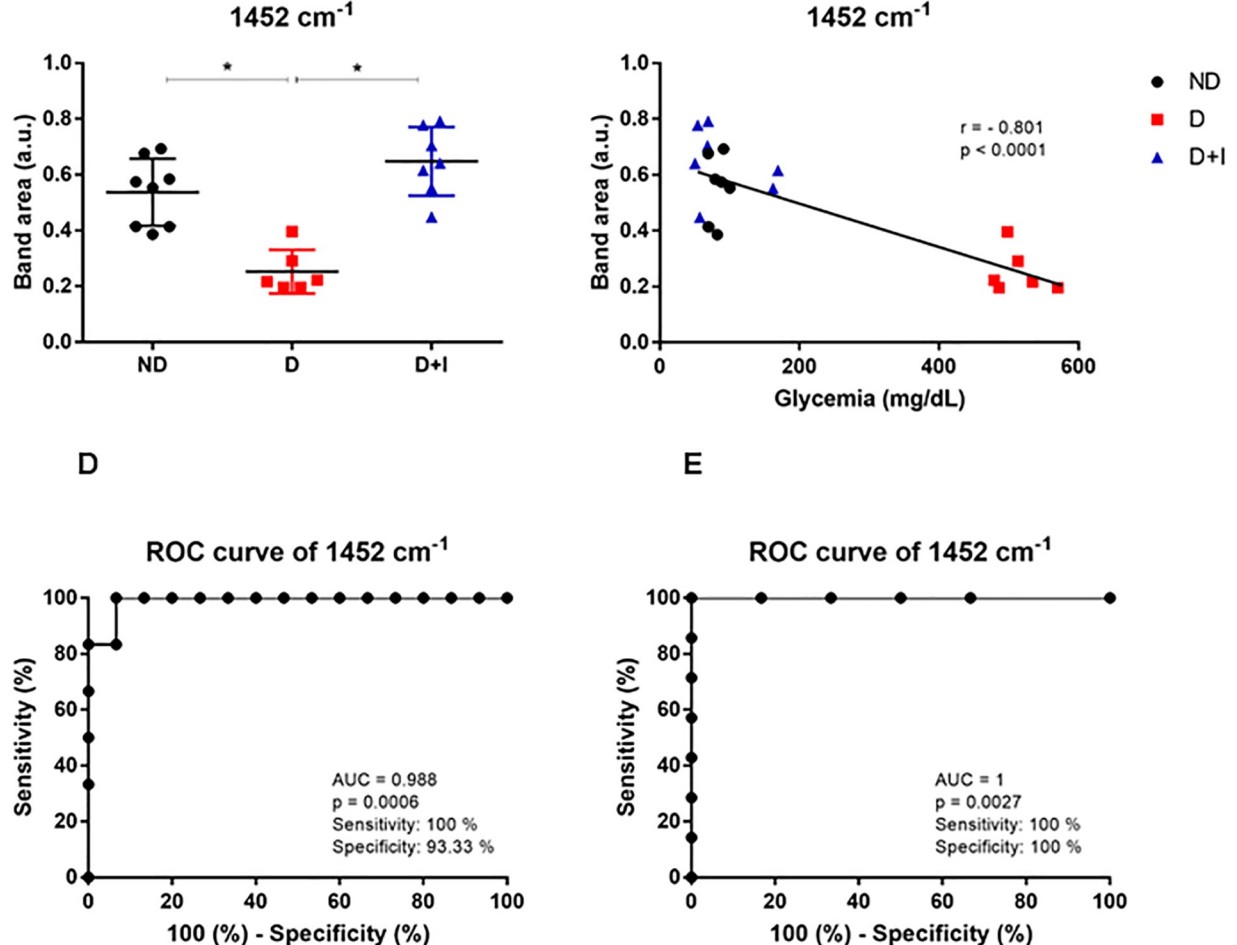

**Fig 2.** Spectral of 1452 cm-1 (A); Band area of 1452 cm-1 (B); Pearson correlation between glycemia and band area of 1452 cm-1 (C); ROC curve analyses of 1452 to normoglycemic and hyperglycemic (D); ROC curve analyses of 1452 to diabetic and diabetic treated with insulin (E). Non-diabetic rats (ND), diabetic rats (D) and diabetic treated with insulin (D+I).

Considering that sensitivity and specificity are basic characteristics to determine the accuracy of diagnostic and monitoring test, ROC curve analysis were used to evaluate the potential diagnostic of these spectral bands under two conditions of analysis. The first one, we analyzed the condition of normoglycemic (ND and D+I) with hyperglycemic (D). The cutoff value to 1452 cm$^{-1}$ band was 0.405, and the corresponding sensitivity and specificity were 100% and 93.3%, respectively. In ROC analysis, the area under the curve (AUC) of this band was 0.988 (Fig 2D). To emphasizes our focus on insulin-treated rats, we also showed ROC curve analysis comparing only D+I with D. Both sensitivity and specificity of 1452 cm$^{-1}$ band was 100% with cutoff of 0.422 (p: 0.0027). Both sensitivity and specificity of 836 cm$^{-1}$ band to differentiate normoglycemic (ND and D+I) than hyperglycemic (D) were 100% with cutoff of 0.128 (Fig 3D). As expected, the ROC curve to differentiate insulin-treated diabetic (D+I) than hyperglycemic (D) showed similar data (Fig 3E).

## Differentiation among the groups by principal component analysis followed by linear discriminant analysis (PCA-LDA) and Hierarchical Cluster Analysis (HCA)

Principal component analysis followed by linear discriminant analysis (PCA-LDA) was performed to reduce the dimensionality of the data set, with the preservation of the variance to evaluate the discrimination between the samples. PCA was performed using 6 principal components (PCs), accounting for 95.2% (20/21) of cumulative variance of correct classification with cross validation. The PCA model considered 95.8% of the data of the spectrum through the second derivative for analyze. The PC1 to PC6 proportions of the spectra variability in the covariance matrix were, 39.0%, 32.0%, 11.3%, 8.2%, 3.1% and 2.2%, respectively. After linear discriminant analysis (LDA) with leave-one-out cross-validation, three groups (ND, D and D+I) were formed, but only one sample belonging to class D+I was classified for group D (Fig 4). S1–S3 Tables show the mean quadratic distance, discriminant linear function and the summary of classification of each sample (with quadratic distance of each sample, prediction, validation and probability), respectively, in saliva of ND, D and D+I rats.

Hierarchical cluster analysis (HCA) was performed to investigate the effects of treatment with insulin on diabetic to the differentiation of non-diabetic and diabetic samples. HCA was performed in part of salivary spectrum. The deconvolution analyzes were done in the five spectral regions represented in Fig 5, as A region (2995 cm$^{-1}$ to 2889 cm$^{-1}$), B region (1664 cm$^{-1}$ to 1581 cm$^{-1}$), C region (1410 cm$^{-1}$ to 1234 cm$^{-1}$), D region (1149 cm$^{-1}$ the 1080 cm$^{-1}$) and E region (1018 cm$^{-1}$ to 955 cm$^{-1}$) which allowed the differentiation of the non-diabetic, diabetic and insulin-treated diabetic. As seen from the Fig 5, all non-diabetics and diabetics were separate with 100% of discrimination. Only one insulin-treated diabetic was categorized as non-diabetic. The total accuracy, which is highly important for potential monitoring applications, was 95.2% (20/21) in HCA analysis.

## Discussion

The development of a novel, rapid, noninvasive tool for the diagnosis, and the most important, for monitoring diabetes mellitus based on the comprehensive analysis of spectral salivary constituents would be of great use to health clinics. Herein, we have investigated the translational applicability of ATR-FTIR spectroscopy with the potential monitoring of metabolic control in

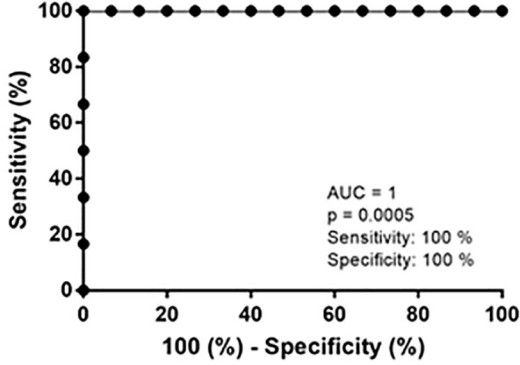

**Fig 3.** Spectral of 836 cm-1 (A); Band area of 836 cm-1 (B); Pearson correlation between glycemia and band area of 836 cm-1 (C); ROC curve analyses of 836 to normoglycemic and hyperglycemic (D); ROC curve analyses of 836 to diabetic and diabetic treated with insulin (E). Non-diabetic rats (ND), diabetic rats (D) and diabetic treated with insulin (D+I).

diabetes. ATR-FTIR detected six potential spectral bands, and, from these, two bands were showed a strong correlation with glycemia and high sensibility and specificity to differentiate hyperglycemic than normoglycemic conditions indicating potential monitoring applicability for diabetes. The discriminatory power of these two salivary ATR-FTIR bands area are candidates for monitoring diabetes under insulin therapy.

As expected in the diabetic state, plasma glucose, urine volume, and urine glucose concentration are increased in non-treated diabetic rats compared to non-diabetic rats [24]. In addition, insulin treatment decreased glycemia, urine volume, and urine glucose. These findings are consistent with other studies [25–28]. It is known that salivary composition changes in diabetes mellitus [29–31]. Also, diabetes mellitus frequently decreases salivary flow, alters the expression of salivary proteins, and increases glucose levels in saliva [29, 31, 32]. From these parameters, it is possible to use salivary components to reflect the presence and severity of hyperglycemia [33]. The saliva of diabetics with poor metabolic control shows an increase in salivary glucose concentration [34]. The correlation of glycemia with glucose concentration in saliva is still not well established, so currently it is not used to verify the degree of metabolic control and diagnosis in diabetes mellitus [35–37]. ATR-FTIR spectroscopy has been used as an alternative discriminatory method to others chronic diseases, due to its major advantages of being label-free and non-destructive, rapid, high-throughput, not requiring sample preparation, and cost-effective analytical method for providing details of the chemical composition and molecular structures [38, 39].

The spectral analysis method to dried saliva described in the present study may be used in rodent and human models. Spectral parameters, such as shifts in band positions and changes in spectral modes intensity, can be used to obtain valuable information about sample composition, which may have diagnostic and monitoring the potential for many diseases [20]. To get relevant information about the concentration of the salivary molecules, integrated band area analysis was performed in the saliva spectra since, according to the Beer-Lambert law, absorption band intensity/band area is proportional to the concentration of the sample [39, 40]. Therefore, differences in the band area for asymmetric $CH_3$ bending modes of the methyl groups of protein (1452 cm$^{-1}$) and $C_2$ endo/anti-B-form helix conformation (836 cm$^{-1}$) differ in salivary constituents among the groups. Bencharit, Baxter [40] showed the differences in the composition of salivary proteins associated with metabolic control in diabetes on a proteomic analysis, and similar quantitative differences were found in the present study analyzed with spectroscopy ATR-FTIR. Type 2 diabetes mellitus induced changes in the lipid and protein components on the erythrocyte membrane and causing structural changes by FTIR spectroscopy in the protein secondary structure with a shift in the beta-sheet and beta-turn structures [41].

These two salivary spectral modes showed a high and significant correlation with metabolic control. Clinically, the most interesting comparisons are the correlation between these salivary spectral band areas and glycemia. Together, these salivary spectral bands showed a 100% sensitivity and 100% specificity in ROC analysis. ROC curve analysis is widely considered to be the most objective and statistically valid method for biomarker performance evaluation [42]. Regarding the potential for translation to the clinic, our results suggest that two salivary band areas, 1452 cm$^{-1}$ and 836 cm$^{-1}$, can be considered noninvasive spectral biomarkers of monitoring diabetes treated with insulin. Different drug treatments and several levels of glycemia should ideally be possible to differentiate; therefore more studies need to be investigated.

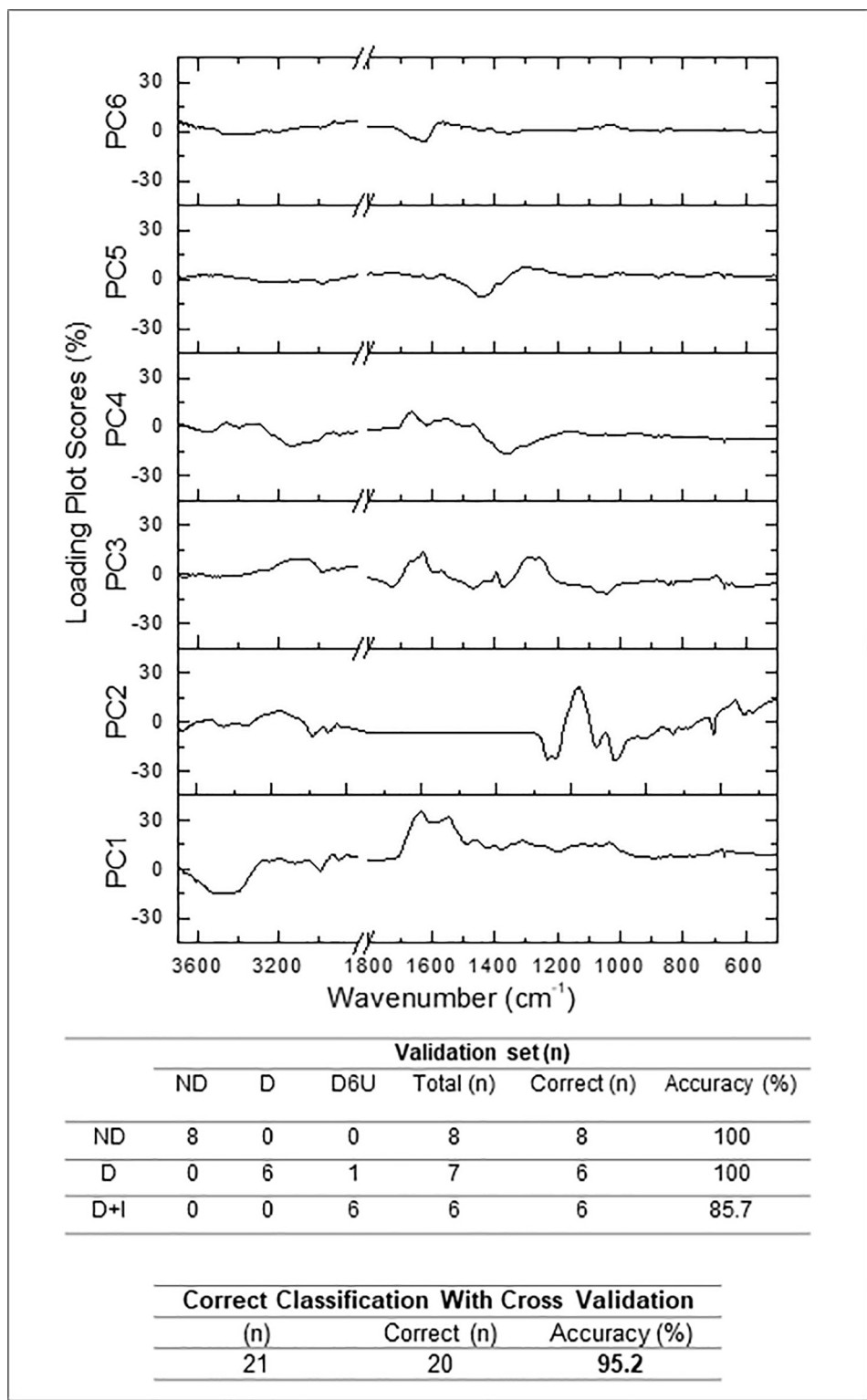

**Fig 4. PCA-LDA analyses.** Non-diabetic rats (ND), diabetic rats (D) and diabetic treated with insulin (D+I).

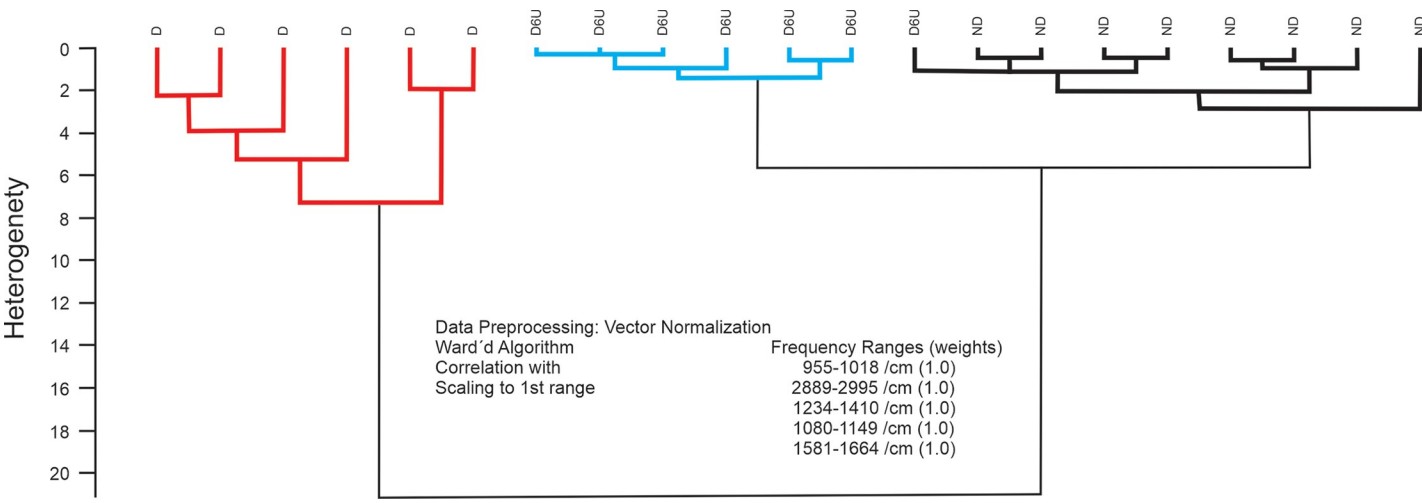

**Fig 5. HCA analyses.** Non-diabetic rats (ND), diabetic rats (D) and diabetic treated with insulin (D+I).

These results indicate that these spectral modes can be used as a diagnostic and monitoring platform for diabetes mellitus; once interestingly, insulin treatment was also able to revert the salivary spectra observed in the hyperglycemic state. Therefore, insulin treatment is not a potential confounding factor that may influence salivary vibrational mode in comparison with glycemia. Some studies have indicated specific salivary biomarkers for diabetes, such as glucose, alpha-amylase, immunoglobulins, myeloperoxidases [9, 30, 43, 44] with similar potential, but not with a focus on disease monitoring and/or with the use of IR spectroscopy. As expected, the commercialization of saliva glucose biosensors has not been used for diabetes management, and new strategies need to be developed to measure salivary components that reflects glycemia. Besides, it is essential bearing in mind that $C_2$ conformation of sugars at 836 $cm^{-1}$ do not indicate a presence of glucose, the aldehyde structure for glucose into a cyclic hemiacetal (glucopyranose) occur in C4-C5 bond brings [45] at 1375 $cm^{-1}$ [46].

Multivariate analysis as PCA-LDA and HCA can be used to discriminate samples based on their spectrum. In FTIR analysis, the diagnostic accuracy for diabetes detection using saliva was 100.0% for the training set and 88.2% for the test (validation) set using linear discriminant analysis (LDA) calculations [22]. However, in the present study, both PCA-LDA and HCA obtained 95.2% of accuracy using saliva to discriminate normoglycemic, diabetic, and insulin-treatment diabetic models. It is essential to emphasize that our protocol used ultra-low values of saliva (2 μl) under airflow dried during only 2 minutes and the other study [22] used 50 μl (25 times higher) under dried during ~30 min at 25 Torr on 13 mm BaF windows. The analysis using univariate analysis was performed only in the present study. Besides, the Pearson's correlation between 1452 $cm^{-1}$ and 836 $cm^{-1}$ vibrational modes with glycemia described in present study showed higher correlation values (r = 0.801 and r = -0.788) comparing with another study ([22]; r = 0.49) using a SCN band, a classical indicator of tobacco smoking (a condition present in ~60% healthy and diabetic subjects).

Cluster analyses confirm its potential to discriminate ND, D, and D+I groups with high accuracy. The success rate for ND e D was 100%, and for D+I was 85.7%. Altogether, the data performed an accuracy of 95.23%. The inclusion of one sample of D+I animals in the non-diabetic control group is expected, considering that insulin is a gold-standard treatment of diabetes. We believe that this infrared analysis opens perspectives to use saliva to monitor the metabolic control with molecules different than glucose. It is unequivocal that glucose is the

main molecule to analyses metabolic control in the blood; however, the demonstration of glucose transporters in the luminal membrane of ductal cells in salivary glands [28] highlight the need to evaluate other biomarkers in saliva.

Although we have shown that ATR-FTIR technology is useful for the identification of possible biomarkers for monitoring diabetes mellitus in the saliva of rats, this is a first exploratory study using ATR-FTIR technology for this purpose. Therefore, further studies are needed to validate the suggested spectral biomarkers in humans and to determine the applicability of this technique for the monitoring of diabetes mellitus in human saliva. As other molecular techniques, the ATR-FTIR can detect functional groups present in several components, which leads to the analysis focus in the intensity of each vibrational mode or multivariate analysis over the detection of a specific type of protein/sugar. It is essential to emphasize that ATR-FTIR have been used for biofluids analysis, allowing same-day detection and grading of a range of diseases in humans [21, 47–51]. Also, one limitation of this study is the inclusion of rats in higher levels of glycemia, which was not intentional but could be explained by the effect of streptozotocin on beta cells. Although requiring further confirmation to provide that our platform is suitable to detect glycemic fluctuation (minutes/hour), the present data indicate that our novel noninvasive approach to diabetes monitoring has the potential to provide discrimination of short-time insulin treatment. Supposing that a similar vibrational mode can discriminate against other conditions than hyperglycemia, we can assume that multivariate chemometric analysis is suitable to discriminate between different diseases. The prospect of identifying spectral biomarkers in saliva open new perspectives for monitoring the severity of diabetes, and compliance with insulin treatment modalities. Considering the similarity of metabolic mechanism between the diabetic hyperglycemic animal's models and diabetic patients, we believe that this salivary ATR-FTIR-based diagnostics could be tested in large samples patients to rapidly and inexpensively monitoring diabetes using saliva samples and even open the possibility for point-of-care assays by portable attenuated total reflectance infrared spectroscopic approaches.

In conclusion, we showed that ATR-FTIR spectroscopy in the saliva could differentiate diabetic from non-diabetic and insulin-treated diabetic rats. Our data suggest specific fingerprint regions (highlighted two salivary spectral modes 1452 cm$^{-1}$ and 836 cm$^{-1}$) capable of discriminating between hyperglycemic and normoglycemic conditions (insulin-treated or not) in univariate analysis. A very high discriminatory accuracy of 95.2% was also obtained for classifying infrared spectra of saliva between diabetic, non-diabetic, and insulin-treated rats by the PCA-LDA and HCA multivariate models. In summary, these salivary results indicate that ATR-FTIR spectroscopy coupled with univariate or multivariate chemometric analysis has the potential to provide a novel noninvasive approach to diabetes monitoring assisting medical decision making to avoid under-treatment or over-treatment with insulin.

## Methods

### Animals

This study was carried out in accordance with recommendations in the Guide for the Care and Use of Laboratory Animals of the Brazilian Society of Laboratory Animals Science (SBCAL). All experimental procedures for the handling, use and euthanasia were approved by the Ethics Committee for Animal Research of the Federal University of Uberlandia (UFU) (License #CEUA-UFU No. 013/2016) according to Ethical Principles adopted by the Brazilian College of Animal Experimentation (COBEA). All effort was taken to minimize the number of animals used and their discomfort.

Male *Wistar* rats (~250g) were obtained from Center for Bioterism and Experimentation at the Federal University of Uberlandia. The animals were maintained under standard conditions

(22 ± 1°C, 60% ± 5% humidity and 12-hour light/dark cycles, light on at 7 AM) and were allowed with free access to standard diet and water at the Institute of Biomedical Sciences rodent housing facility.

### Induction of diabetes and insulin treatment

Diabetes was induced in overnight-fasted animals by an intraperitoneal injection (60 mg/kg) of streptozotocin (STZ) (Sigma-Aldrich, St. Louis, MO. USA) dissolved in 0.1 M citrate buffer (pH 4.5). Animals with hyperglycemia (>250 mg/dl) were chosen as diabetics. Non-diabetic animals received injection of NaCl 0.9% in similar volume. Twenty one days later after induction of diabetes, diabetic rats were submitted to a 7-day treatment with vehicle (ND and D) or with 6U of insulin [NPH insulin, Biohulin N; Biobrás, MG, Brazil] (D+I) per day (2U at 8:30 a.m. and 4U at 5:30 p.m.) subcutaneously [26]. Thus, animals were divided in Non-Diabetic (ND, n = 8), Diabetic (D, n = 6) and diabetic treated with 6U insulin (D+I, n = 7). Glucose levels in overnight-fasted were obtained from the tail vein and measured using reactive strips (Accu-Chek Performa, Roche Diagnostic Systems, Basel, Switzerland) by a glucometer (Accu-Chek Performa, Roche Diagnostic Systems, Basel, Switzerland) in the moment of samples collection.

In the last day of treatment, the animals were kept in metabolic cages and water intake, food intake, urine volume were measured. Urine was collected over 24 h and the glucose concentration in the urine was evaluated using an enzymatic Kit (Labtest Diagnostica SA, Brazil). Besides that, variation of gain/loss body weight (Δ body weight) compared parameters in STZ or vehicle induction with parameters after insulin or vehicle treatment.

### Saliva collection

After 7-days of treatment, the animals were anaesthetized by an intraperitoneal injection with ketamine (100 mg/kg) and xylazine (20 mg/kg). Stimulated saliva was collected with parasympathetic stimulation through pilocarpine injection (2 mg/kg, i.p.). Stimulated saliva was collected in pre weighed flasks for 10 min from the oral cavity [28]. The collected saliva was stored at -80°C for further processing and analysis. The animals were euthanized with excessive anesthetic dose, after samples collection.

### Chemical profile in stimulated saliva by ATR-FTIR spectroscopy

Salivary spectra were recorded in 3000 cm$^{-1}$ to 400 cm$^{-1}$ region using ATR-FTIR spectrophotometer Vertex 70 (Bruker Optics, Reinstetten, Germany) using a micro-attenuated total reflectance (ATR) component. The crystal material in ATR unit was a diamond disc as internal-reflection element. The salivary pellicle penetration depth ranges between 0.1 and 2 μm and depends on the wavelength, incidence angle of the beam and the refractive index of ATR-crystal material. In the ATR-crystal the infrared beam is reflected at the interface toward the sample. Saliva was directly dried using airflow on ATR-crystal for 2 min before salivary spectra recorded. The air spectra was used as a background in ATR-FTIR analysis. Sample spectra and background was taken with 4 cm$^{-1}$ of resolution and 32 scans were performed for salivary analysis [52, 53].

### Spectra data evaluation procedures

The spectra data obtained were processed using Opus 6.5 software (Bruker Optics, Reinstetten, Germany). Measurements were performed in mid-infrared region (3000–400 cm$^{-1}$) with spectral resolution of 4 cm$^{-1}$ and 32 scans per spectrum. Samples were pressed into ATR diamond crystal with standardized pressure. For the generation of mean spectra and band areas, the

spectra were normalized by vector and baseline corrected to avoid errors during the sample preparations and spectra analysis. To evaluate the mean values for the peak positions, band area of the spectra was considered belonging to each animal of the groups. The band positions were measured using the frequency corresponding to the center of weight of each band. Band areas were calculated from normalized and baseline corrected spectra using OPUS software. Sensitivity and specificity values were calculated based on the external test set as follows:

The specificity or true negative rate is defined as the percentage of rats who are correctly identified as being normoglycemic Non-Diabetic (ND) or normoglycemic D+I:

$$\text{Specificity} = \frac{TN}{TN + FP}$$

The quantity 1-specificity is the false positive rate and is the percentage of rats that are incorrectly identified as diabetic (D).

The sensitivity or true positive rate is defined as the percentage of rats who are correctly identified as diabetic (D):

$$\text{Sensitivity} = \frac{TP}{TP + FN}$$

where TP stands for true positives; TN for true negatives; FP for false positives; and FN for false negatives [54].

## Principal component analysis followed by linear discriminant analysis (PCA-LDA) and Hierarchical Cluster Analysis (HCA)

The principal components analysis (PCA) was used as one step before Linear Discriminant Analysis (LDA) to avoid multicollinearity (Jolliffe & Cadima *et al.*, 2016)] performed by Minitab® Program. In order to calculate the principal components, the data was normalized and then matrix of spectra was centered, which was the mean spectrum was subtracted from every spectrum in the matrix. This procedure removes the redundancy of the average in the dataset interpretation. The principal components (PC) were calculated using a full range of the FT-IR spectra (ND, D and D+I) between 3700 and 500 cm-1, and a covariance matrix, where the original variables are reduced to the most important descriptive components. The original number of PC is always equal the number of variables and in this study, the first six principal components (PC1-PC6) were used to calculate the LDA that corresponds to 95,8% of the cumulative proportion of the spectrum variability. The LDA with leave-one-out cross validation [55] was done according to the pathological reports.

Infrared spectra of saliva samples were also analyzed by OPUS software (version 4.2) using HCA. In the first step, the vector normalization was performed calculating the average absorbance (y axis) value of the selected spectra regions and subtracted this value from the spectrum, which technically centered around 0. After this procedure OPUS calculates the sum of squares of all y value, and the respective spectrum is divided by the root this sum. The Scaling to 1st Range method determined the minimum and maximum value of spectral distances for the first spectral range. The dendrogram was performed by Ward's clustering algorithm in the defined spectral regions determines the growth of heterogeneity H, merging all homogeneous spectra into a group using OPUS User manual.

## Statistical analysis

The data of the band area were analyzed using the one-way analysis of variance (ANOVA), followed by Tukey Multiple Comparison as a *post-hoc* test. The correlation between values of

blood glucose concentration and salivary band areas of the spectra were analyzed by the Pearson correlation test. For all spectral band candidates, we constructed the Receiver Operating Characteristic (ROC) curve and computed the area under the curve (AUC) value, sensitivity and specificity by numerical integration of the ROC curve. The Kolmogorov-Smirnov test was applied to test the normality of the variables. All these analyses were performed using the software GraphPad Prism (GraphPad Prism version 7.00 for Windows, GraphPad Software, San Diego, CA, USA). Only values of $p < 0.05$ were considered significant and the results were expressed as mean ± S.D.

## Supporting information

**S1 Fig.** Spectral of 2924 cm-1 (A); Band area of 2924 cm-1 (B); Pearson correlation between glycemia and band area of 2924 cm-1 (C); ROC curve analyses of 2924 cm-1 to normoglycemic and hyperglycemic (D); ROC curve analyses of 2924 cm-1 to diabetic and diabetic treated with insulin (E). Non-diabetic rats (ND), diabetic rats (D) and diabetic treated with insulin (D+I). (TIF)

**S2 Fig.** Spectral of 1549 cm-1 (A); Band area of 1549 cm-1 (B); Pearson correlation between glycemia and band area of 1549 cm-1 (C); ROC curve analyses of 1549 cm-1 to normoglycemic and hyperglycemic (D); ROC curve analyses of 1549cm-1 to diabetic and diabetic treated with insulin (E). Non-diabetic rats (ND), diabetic rats (D) and diabetic treated with insulin (D+I). (TIF)

**S3 Fig.** Spectral of 1313 cm-1 (A); Band area of 1313 cm-1 (B); Pearson correlation between glycemia and band area of 1313 cm-1 (C); ROC curve analyses of 1313 cm-1 to normoglycemic and hyperglycemic (D); ROC curve analyses of 1313 cm-1 to diabetic and diabetic treated with insulin (E). Non-diabetic rats (ND), diabetic rats (D) and diabetic treated with insulin (D+I). (TIF)

**S4 Fig.** Spectral of 1120 cm-1 (A); Band area of 1120 cm-1 (B); Pearson correlation between glycemia and band area of 1120 cm-1 (C); ROC curve analyses of 1120 cm-1 to normoglycemic and hyperglycemic (D); ROC curve analyses of 1120 cm-1 to diabetic and diabetic treated with insulin (E). Non-diabetic rats (ND), diabetic rats (D) and diabetic treated with insulin (D+I). (TIF)

**S1 Table. Mean quadratic distance in saliva of ND, D and D+I rats.**
(DOCX)

**S2 Table. Discriminant linear function in saliva of ND, D and D+I rats.**
(DOCX)

**S3 Table. Summary of classification with the quadratic distance of each sample, prediction, validation and probability of each sample in saliva of ND, D and D+I rats.**
(DOCX)

## Author Contributions

**Conceptualization:** Douglas C. Caixeta, Emília M. G. Aguiar, Léia Cardoso-Sousa, Foued S. Espindola, Leandro Raniero, Walter L. Siqueira, Robinson Sabino-Silva.

**Data curation:** Douglas C. Caixeta, Léia Cardoso-Sousa, Líris M. D. Coelho, Stephanie W. Oliveira.

**Formal analysis:** Douglas C. Caixeta, Emília M. G. Aguiar, Léia Cardoso-Sousa, Líris M. D. Coelho, Stephanie W. Oliveira, Leandro Raniero, Karla T. B. Crosara.

**Funding acquisition:** Foued S. Espindola.

**Investigation:** Douglas C. Caixeta, Emília M. G. Aguiar, Foued S. Espindola, Leandro Raniero, Karla T. B. Crosara, Walter L. Siqueira, Robinson Sabino-Silva.

**Methodology:** Douglas C. Caixeta, Emília M. G. Aguiar, Léia Cardoso-Sousa, Líris M. D. Coelho, Stephanie W. Oliveira, Foued S. Espindola, Leandro Raniero, Karla T. B. Crosara, Walter L. Siqueira, Robinson Sabino-Silva.

**Project administration:** Robinson Sabino-Silva.

**Resources:** Douglas C. Caixeta, Robinson Sabino-Silva.

**Software:** Matthew J. Baker.

**Supervision:** Douglas C. Caixeta, Foued S. Espindola, Leandro Raniero, Karla T. B. Crosara, Matthew J. Baker, Walter L. Siqueira, Robinson Sabino-Silva.

**Validation:** Douglas C. Caixeta, Leandro Raniero, Matthew J. Baker, Walter L. Siqueira, Robinson Sabino-Silva.

**Visualization:** Douglas C. Caixeta, Emília M. G. Aguiar, Stephanie W. Oliveira, Leandro Raniero, Karla T. B. Crosara, Matthew J. Baker, Walter L. Siqueira, Robinson Sabino-Silva.

**Writing – original draft:** Douglas C. Caixeta, Emília M. G. Aguiar, Léia Cardoso-Sousa, Foued S. Espindola, Leandro Raniero, Karla T. B. Crosara, Matthew J. Baker, Walter L. Siqueira, Robinson Sabino-Silva.

**Writing – review & editing:** Douglas C. Caixeta, Emília M. G. Aguiar, Léia Cardoso-Sousa, Líris M. D. Coelho, Stephanie W. Oliveira, Foued S. Espindola, Leandro Raniero, Karla T. B. Crosara, Matthew J. Baker, Walter L. Siqueira, Robinson Sabino-Silva.

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
