## [Decision Letter · Decision Letter 0]

9 Dec 2019

PONE-D-19-26190

Salivary molecular spectroscopy: a rapid and non-invasive monitoring tool for diabetes mellitus during insulin treatment

PLOS ONE

Dear Dr. Sabino-Silva,

Thank you for submitting your manuscript to PLOS ONE. After careful consideration, we feel that it has merit but does not fully meet PLOS ONE’s publication criteria as it currently stands. Therefore, we invite you to submit a revised version of the manuscript that addresses the points raised during the review process.

The reviewers came up with a list of matters and suggestions, which should carefully be considered in the revision. In principle, I fully agree with the reviewers believing your work is of interest and should be published.

Nevertheless, I feel that the stage of your work and its limitations should be better pinpointed. It appears to be a noteworthy starting point, but not yet at the stage of justifying claims about usefulness in diabetes monitoring. As far as I understand, you do neither know, how far your readouts reflect acute hyperglycaemia or long-term metabolic derangement, nor if a similar signature in saliva would likewise arise from other conditions than hyperglycaemia/diabetes. If possible, please provide more information on these aspects (including the request from the reviewers about which chemical entities the peaks represent, e.g. by "spiking" the samples). If there is no additional information available, these limitations and obstacles on the way to the clinic should be openly addressed and the impression of premature claims should be avoided by careful wording.

We would appreciate receiving your revised manuscript by Jan 23 2020 11:59PM. To enhance the reproducibility of your results, we recommend that if applicable you deposit your laboratory protocols in protocols.io, where a protocol can be assigned its own identifier (DOI) such that it can be cited independently in the future. For instructions see: http://journals.plos.org/plosone/s/submission-guidelines#loc-laboratory-protocols

We look forward to receiving your revised manuscript.

Kind regards,

Clemens Fürnsinn, Ph.D.

Academic Editor

PLOS ONE

Journal Requirements:

1.

2. To comply with PLOS ONE submissions requirements, please provide methods of sacrifice in the Methods section of your manuscript.

Reviewers' comments:

Reviewer's Responses to Questions

**Comments to the Author**

1. Is the manuscript technically sound, and do the data support the conclusions?

Reviewer #1: Yes

Reviewer #2: Partly

Reviewer #3: Partly

2. Has the statistical analysis been performed appropriately and rigorously? 

Reviewer #1: Yes

Reviewer #2: I Don't Know

Reviewer #3: N/A

3. Have the authors made all data underlying the findings in their manuscript fully available?

Reviewer #1: Yes

Reviewer #2: Yes

Reviewer #3: No

4. Is the manuscript presented in an intelligible fashion and written in standard English?

Reviewer #1: Yes

Reviewer #2: Yes

Reviewer #3: No

5. Review Comments to the Author

Reviewer #1: This is an interesting paper presenting an analysis of a new and simple IR-based method to distinguish among states of hyperglycemia. The presentation is reasonably clear and the methodology seems sound. I only have minor comments:

1. The description of insulin-treated animals as D6U is unnecessarily confusing. These are insulin-treated, no longer hyperglycemic; the units matter for methods but not for the group descriptor. Suggest these be designated as D+I or some comparably easy and more relevant descriptor

2. The actual method for insulin treatment should be augmented to include the type of insulin. Why these particular doses (rather than something based on weight or glycemia for example?)

3. I don't find the continuous analyses of signal AUC against glucose very compelling. This study is set up to provide large group differences, and these come out as clusters of the data with a pseudo-line relating the variables. The more direct presentation of the comparison is what is done with the box/whisker plots.

4. It is of interest whether these peaks are measuring glucose or a directly related carbohydrate, or something entirely separate. Is that known?

5. I really like the detailed mathematical analyses using multidimensional methods, adds considerably to the paper

6. It is obvious that this now needs to be further tested in humans with various stages of diabetes, and as a potential marker of actual glucose values. Can a method be developed where a portable device can perform the physical measure and undertake computations to give a useful and actionable output? (I realize this is the 'next' project but it seems the logical extension; perhaps a comment in this paper could be added about challenges of moving this into diagnostic or clinical use)

7. The English is overall very good but occasionally there are non-standard English phrases (and some grammatical issues). Suggest this paper be passed by a native English speaker in its final version

Reviewer #2: Have not reviewed yet.

Please see my confidential note to the editor.

Have not reviewed yet.

Please see my confidential note to the editor.

Have not reviewed yet.

Please see my confidential note to the editor.

Reviewer #3: Comments to the authors

The authors describe an FTIR-based method to detect spectral changes in the saliva of diabetic mice and compare them with control and treated diabetic mice. The study presents some interesting results which need further biochemical validation, in order to verify that the observed spectral changes are useful for glucose monitoring in the saliva.

Major comments.

1. The main finding of the authors is the observed spectral decrease at 1452 and 836 cm-1. However, it is not clear what these peaks represent. According to the authors, the 1452 peak corresponds to methyl groups of proteins and the 836 groups to sugars. The authors should provide some kind of evidence on which proteins absorb at 1452 and why they are decreased in the diabetic mice. Furthermore, is the 836 a specific signature of glucose in the saliva? If yes, this peak should be increased in diabetic mice that have higher glucose concentrations in their blood.

2. The changes in the spectra of diabetic mice are subtle and measured in arbitrary units. Therefore, they are not convincing. The authors should show the FTIR spectrum of the diabetic mice after having removed as a background either the spectrum of non-diabetic or insulin-treated mice. Then, they would indicate spectral changes as variations from the respective baseline.

3. Would the authors detect the 1452 and 836 peak changes also in human samples from diabetic patients?

4. The paper would benefit if the authors could e.g. measure the FTIR spectra of increasing concentrations of glucose solutions and compare them with the spectra obtained from the 3 different mice groups.

5. In the discussion section, the authors describe that glucose concentration can be measured in the saliva of diabetic individuals. Therefore, the authors should validate the presence of glucose in their saliva samples and measure its concentration. The biochemical validation of their results is missing.

Minor comments

1. The values for the reduced weight gain in Table 1 are misleading. For example, it shows a -2.7 change with a SD of 11.3. Instead, the authors should consider to show the average body weight and its SD. Also, they should clearly state what the * and # symbols indicate.

2. Readers that are unfamiliar with ROC curves may not easily understand the rationale of these statistical tests and their downstream analysis. The authors should consider to rephrase lines 172-184. Furthermore, usually ROC curves show a (1-specificity) in their x-axis. Here, the authors show specificity in their ROC curves. Is this correct? Also, does each point in these curves represent a single animal? The equal distance between data points seems unusual.

3. The last two sentences of the results section (lines 208-209) are the conclusion of the PCA analysis. The authors should move them at the end of the previous paragraph.

4. The authors should consider to shorten the Discussion section. At its present form, it seems very long compared to the other sections of the paper

5. Description of the animal groups is misleading. The author should state that they originally induced diabetes in 13 animals and then divided them in the D and D6U groups. The first sentence in Line 343 should move after the selection of mice with hyperglycemia.

6. The authors should use different colors for the marks (animals of each group) in Figure 2C and all other similar figures. Could they also use similar colors for the marks in parts D-E?

7. The authors should check the legends of Figure 3, 4, S1-S4. Some words seem to be missing. For example, line 716, ROC curve analysis of 836 cm-1 to diabetic mice and diabetic mice treated with insulin.

8. In Figure 4, the authors should state the % proportion of variance of analyzed data in the y-axis.

6. PLOS authors have the option to publish the peer review history of their article (what does this mean?). If published, this will include your full peer review and any attached files.

Reviewer #1: No

Reviewer #2: Yes: Dr Ylias Sabri

Reviewer #3: No

---

## [Author Response · Author response to Decision Letter 0]

27 Jan 2020

The rebuttal letter was attached previously.

---

## [Decision Letter · Decision Letter 1]

5 Feb 2020

PONE-D-19-26190R1

Salivary molecular spectroscopy: a sustainable, rapid and non-invasive monitoring tool for diabetes mellitus during insulin treatment

PLOS ONE

Dear Dr. Sabino-Silva,

Thank you for submitting your manuscript to PLOS ONE. After careful consideration, we feel that it has merit but does not fully meet PLOS ONE’s publication criteria as it currently stands. Therefore, we invite you to submit a revised version of the manuscript that addresses the points raised during the review process.

I am in principle inclined to support publication of your work also without additional data, but there is still need for a carefully revised version. As emphasised by the reviewer as well as by myself in the first round of revision, open statements about limitations of your work, about which points are not yet answered, and about the stage of your work on the way to clinical use are more important than frail claims about assumed future usefulness. Obstacles like, e.g., that it seems unclear whether your saliva readout reflects short or long term glucose fluctuations, and if a similar outcome could arise from other conditions than hypergylcaemia are still not addressed. Please clarify these matters and conscientiously follow the comments from the reviewer.

We would appreciate receiving your revised manuscript by Mar 21 2020 11:59PM. To enhance the reproducibility of your results, we recommend that if applicable you deposit your laboratory protocols in protocols.io, where a protocol can be assigned its own identifier (DOI) such that it can be cited independently in the future. For instructions see: http://journals.plos.org/plosone/s/submission-guidelines#loc-laboratory-protocols

We look forward to receiving your revised manuscript.

Kind regards,

Clemens Fürnsinn, Ph.D.

Academic Editor

PLOS ONE

Reviewers' comments:

Reviewer's Responses to Questions

**Comments to the Author**

1. If the authors have adequately addressed your comments raised in a previous round of review and you feel that this manuscript is now acceptable for publication, you may indicate that here to bypass the “Comments to the Author” section, enter your conflict of interest statement in the “Confidential to Editor” section, and submit your "Accept" recommendation.

Reviewer #3: (No Response)

2. Is the manuscript technically sound, and do the data support the conclusions?

Reviewer #3: Yes

3. Has the statistical analysis been performed appropriately and rigorously? 

Reviewer #3: Yes

4. Have the authors made all data underlying the findings in their manuscript fully available?

Reviewer #3: Yes

5. Is the manuscript presented in an intelligible fashion and written in standard English?

Reviewer #3: No

6. Review Comments to the Author

Reviewer #3: I realize that the authors decided not to include any new experimental data, as I would expect, but rather tried to better explain their results in the revised Discussion section. However, within this lengthy section, I don't see a clear statement on what the.g. on their claim that the 1452 spectrum may correspond to several protein components. Mainly, I don't see a clear statement on the limitations of their study, merely the value of their experimental approach.

Also, I have seen several inconsistencies in their rebuttal letter.

1. The authors claim that inserted Lines 172-184 but I don't see any changes there

2. They also state that the manuscript has been thoroughly checked by native English speakers. However, I found more than one mistakes. E.g. "Line 320, it is important emphasizes..., Line 181 To emphasizes our focus...., Line 437 the term DU6 that was deleted is used again.

3. Figure 4. The authors included the symbol % bit no numerical value for the proportionof variance that each PC diagram explains.

My overall impression is that the authors have not carefully prepared their revised manuscript and response to the reviewers. I would encourage them to resubmit a more careful revision of their paper, in which they would also clearly state the limitations and potential obstacles on the way to the clinic.

7. PLOS authors have the option to publish the peer review history of their article (what does this mean?). If published, this will include your full peer review and any attached files.

Reviewer #3: No

---

## [Author Response · Author response to Decision Letter 1]

18 Feb 2020

The rebuttal letter was attached previously.

---

## [Decision Letter · Decision Letter 2]

25 Feb 2020

Salivary molecular spectroscopy: a sustainable, rapid and non-invasive monitoring tool for diabetes mellitus during insulin treatment

PONE-D-19-26190R2

Dear Dr. Sabino-Silva,

We are pleased to inform you that your manuscript has been judged scientifically suitable for publication and will be formally accepted for publication once it complies with all outstanding technical requirements.

With kind regards,

Clemens Fürnsinn, Ph.D.

Academic Editor

PLOS ONE

Additional Editor Comments (optional):

Please correct the minor remaining typos, which the reviewer has spotted:

Line 297 It is essential emphasizes that our protocol used...

Line 311 the main molecule to following metabolic control in the blood;

Line 323 It is essential emphasize

Line 333 is suitable do discriminate

I still think that there are several typos that the authors should correct

Reviewers' comments:

Reviewer's Responses to Questions

**Comments to the Author**

1. If the authors have adequately addressed your comments raised in a previous round of review and you feel that this manuscript is now acceptable for publication, you may indicate that here to bypass the “Comments to the Author” section, enter your conflict of interest statement in the “Confidential to Editor” section, and submit your "Accept" recommendation.

Reviewer #3: (No Response)

2. Is the manuscript technically sound, and do the data support the conclusions?

Reviewer #3: Yes

3. Has the statistical analysis been performed appropriately and rigorously? 

Reviewer #3: Yes

4. Have the authors made all data underlying the findings in their manuscript fully available?

Reviewer #3: Yes

5. Is the manuscript presented in an intelligible fashion and written in standard English?

Reviewer #3: No

6. Review Comments to the Author

Reviewer #3: Line 297 It is essential emphasizes that our protocol used...

Line 311 the main molecule to following metabolic control in the blood;

Line 323 It is essential emphasize

Line 333 is suitable do discriminate

I still think that there are several typos that the authors should correct at least at the proofs stage

7. PLOS authors have the option to publish the peer review history of their article (what does this mean?). If published, this will include your full peer review and any attached files.

Reviewer #3: No

---

## [Editor Report · Acceptance letter]

3 Mar 2020

PONE-D-19-26190R2 

Salivary molecular spectroscopy: a sustainable, rapid and non-invasive monitoring tool for diabetes mellitus during insulin treatment 

Dear Dr. Sabino-Silva:

I am pleased to inform you that your manuscript has been deemed suitable for publication in PLOS ONE. Congratulations! Your manuscript is now with our production department. 

With kind regards,

on behalf of

Prof. Dr. Clemens Fürnsinn 

Academic Editor

PLOS ONE